# Genomic Characterization of Multidrug-Resistant Extended Spectrum β-Lactamase-Producing *Klebsiella pneumoniae* from Clinical Samples of a Tertiary Hospital in South Kivu Province, Eastern Democratic Republic of Congo

**DOI:** 10.3390/microorganisms11020525

**Published:** 2023-02-18

**Authors:** Leonid M. Irenge, Jérôme Ambroise, Bertrand Bearzatto, Jean-François Durant, Maxime Bonjean, Jean-Luc Gala

**Affiliations:** 1Center for Applied Molecular Technologies, Institute of Experimental and Clinical Research, Université Catholique de Louvain (UCLouvain), 1200 Woluwe-Saint-Lambert, Belgium; 2Defence Laboratories Department, ACOS Ops & Trg, Belgian Armed Forces, 1800 Peutie, Belgium

**Keywords:** extended spectrum β-lactamase, *Klebsiella pneumoniae*, phylogeny, antimicrobial resistance (AMR), antibiotic resistance gene (ARG), virulence factors, Democratic Republic of Congo, eastern DRC, whole-genome sequencing, genetic diversity

## Abstract

Multidrug-resistant (MDR) and extended spectrum β-lactamase (ESBL)-producing extra-intestinal *K. pneumoniae* are associated with increased morbidity and mortality. This study aimed to characterize the resistance and virulence profiles of extra-intestinal MDR ESBL-producing *K. pneumoniae* associated with infections at a tertiary hospital in South-Kivu province, DRC. Whole-genome sequencing (WGS) was carried out on 37 *K. pneumoniae* isolates displaying MDR and ESBL-producing phenotype. The assembled genomes were analysed for phylogeny, virulence factors and antimicrobial resistance genes (ARG) determinants. These isolates were compared to sub-Saharan counterparts. *K. pneumoniae* isolates displayed a high genetic variability with up to 16 sequence types (ST). AMR was widespread against β-lactamases (including third and fourth-generation cephalosporins, but not carbapenems), aminoglycosides, ciprofloxacin, tetracycline, erythromycin, nitrofurantoin, and cotrimoxazole. The *bla*_CTX-M-15_ gene was the most common β-lactamase gene among *K. pneumoniae* isolates. No carbapenemase gene was found. ARG for aminoglycosides, quinolones, phenicols, tetracyclines, sulfonamides, nitrofurantoin were widely distributed among the isolates. Nine isolates had the colistin-resistant R256G substitution in the *pmrB* efflux pump gene without displaying reduced susceptibility to colistin. Despite carrying virulence genes, none had hypervirulence genes. Our results highlight the genetic diversity of MDR ESBL-producing *K. pneumoniae* isolates and underscore the importance of monitoring simultaneously the evolution of phenotypic and genotypic AMR in Bukavu and DRC, while calling for caution in administering colistin and carbapenem to patients.

## 1. Introduction

Infections caused by MDR Gram-negative bacteria (GNB) are becoming a major public health issue all over the world. Because of the limited antimicrobial treatment options, these infections are frequently associated with high morbidity, prolonged hospitalization, high healthcare costs, and poor outcome [1]. These bacteria include MDR ESBL-producing *Enterobacteriaceae*, which have been identified as a major contributor to the current AMR crisis by the World Health Organization (WHO) [2]. According to data from Asia, Latin America, Europe, and Africa, the dramatic increase in third-generation cephalosporin resistance in *Enterobacteriaceae* is mainly due to the spread of *bla*_CTX-M-15_ gene ESBL [3,4,5], with *bla*_CTX-M-15_ gene being the most common ESBL gene worldwide [6,7]. While most studies of MDR ESBL-producing *Enterobacteriaceae* have focused on *Escherichia coli* [8,9,10,11,12,13,14,15], few studies have characterized the whole genome of MDR ESBL-producing *Klebsiella pneumoniae* in Africa, even though this agent is one of the most widespread MDR ESBL-producing *Enterobacteriaceae* associated with extra-intestinal infections both in hospital and community settings [16,17,18,19,20,21]. Moreover, several strains of *K. pneumoniae* have been linked to multinational epidemics all over the world [21,22]. In the DRC, there is little epidemiological data on *K. pneumoniae* and its role in the emergence of hospital and community-acquired infections. Nonetheless, the few available studies that have been conducted in the country show an increasing prevalence of MDR *K. pneumoniae* in hospital-acquired infections [16,17,23]. We previously reported a high prevalence of MDR ESBL-producing *E. coli*, *K. pneumoniae*, and *Enterobacter cloacae* in the urinary tract and bloodstream infections in eastern DRC [16,17]. WGS performed later on MDR ESBL-producing ExPEC isolates from the same collection showed the high prevalence of the ST131 and ST405 pandemic clones carrying the plasmid-mediated CTX-M-15 gene as the primary cause of ESBL-production in EXPEC [9]. Accordingly, the purpose of this study was to characterize extra-intestinal EBSL-producing *K. pneumoniae* isolates from the South-Kivu province of DRC using WGS. These isolates were collected in the years 2013 through 2017. The WGS data allowed phylogenomic analyses, as well as the characterization of the virulome and the resistome.

## 2. Materials and Methods

### 2.1. Ethical Clearance

The study protocols were approved by the Ethical Review Board (ERB) of the Université Catholique de Bukavu, DRC (UCB/CIES/NC/001/2023). The ERB granted a waiver of informed consent after deciding that the study complied with international guidelines on antibiotic surveillance studies. Samples were analysed anonymously to protect confidentiality.

### 2.2. Bacterial Strains

Between September 2013 and June 2017, GNB (n = 2051) were isolated from patients who presented to the 385-bed General Hospital of Bukavu (South Kivu province, DRC) with urinary tract infections (n = 1608), bloodstream infections (n = 416), as well as peritonitis (n = 12), pleurisy (n = 7), and soft tissue infections (n = 8). GNB bacteria were isolated and identified at the species level in urine samples as previously described [16]. Likewise, GNB associated with bloodstream, peritoneal, pleural, and soft skin tissue infections were isolated and identified at the species level following the protocol established in 2014 at HGPRB [17]. All the *K. pneumoniae* isolates (n = 449) from this 2013–2017 collection represented 21.9% of all extra-intestinal GNB isolates. They were selected for additional analyses, including antimicrobial susceptibility testing. 

### 2.3. Antimicrobial Susceptibility Testing

*K. pneumoniae* isolates were tested for susceptibility to 14 antimicrobial agents (i.e., amikacin, gentamicin, ampicillin, amoxicillin, amoxicillin/clavulanic acid, cefepime, ceftriaxone, ceftazidime, cefotaxime, ciprofloxacin, imipenem, meropenem, nitrofurantoin, and colistin) by the method of minimum inhibitory concentration (MIC). The MIC for each of the 14 antimicrobial agents minus colistin was determined after 16–20 h incubation on Mueller–Hinton plates inoculated with suspensions of isolates at a fixed density (0.5 to 0.6 McFarland standard), using MIC test strips (Liofilchem, Roseto degli Abruzzi, Italy) according to the manufacturer’s recommendations. MIC for colistin was determined by the broth microdilution method, using the ComASP^®^ Colistin 0.25–16 µg/mL panel (Liofilchem, Roseto degli Abruzzi, Italy) according to the manufacturer’s recommendations. The European Committee on Antimicrobial Susceptibility Testing (EUCAST) breakpoints, which were updated in January 2022, were used to interpret the susceptibility or resistance status of each isolate. ESBL-production assay was performed by using the MIC Test Strip ESBL Ceftazidime (CAZ)/Ceftazidime + Clavulanic acid (CAL) panel (Liofilchem, Roseto degli Abruzzi, Italy) according to the manufacturer’s recommendations). *E. coli* ATCC 35218 and *K. pneumoniae* ATCC 700603 strains were used as ESBL-negative and positive controls, respectively, while *E. coli* ATCC 25922 and *E. coli* NCTC 13846 were used as colistin-susceptible and resistant strains, respectively.

### 2.4. Bacterial DNA Extraction

*K. pneumoniae* isolates were cultured overnight in 10 mL Luria-Bertani broth. Genomic DNA was isolated using the EZ1 Advanced XL Biorobot and the tissue DNA kit (Qiagen, Hilden, Germany) with the Bacterial card, according to the manufacturer’s instructions. For each isolate, genomic DNA was quantified using Qubit^®^ fluorometric quantitation (ThermoFisher Scientific, Hillsboro, OR, USA) and normalized to 0.2 ng/μL.

### 2.5. Library Preparation and Next Generation Sequencing

The library was prepared using the Nextera XT DNA Library preparation kit (Illumina, San Diego, CA, USA) as previously described [12]. Briefly, bacterial DNA was fragmented, and then tagged with sequencing adapters using the Nextera transposome (Nextera XT DNA Library Preparation Kit, Illumina, San Diego, CA, USA). PCR reaction of 12 cycles was performed to amplify the DNA fragments and to incorporate indices for dual-indexed sequencing of pooled libraries. The 1 nM pooled library was denaturated and diluted prior to loading on a MiSeq paired-end 2 × 150 (MiSeq reagent kit V2 (300 cycles) or 2 × 300 base pairs (bp) (MiSeq reagent kit V3 (600 cycles) sequence run.

### 2.6. Genome Assembly, Annotation, and Alignment

Raw sequencing data from all *K. pneumoniae* isolates were submitted to the European Nucleotide Archive (ENA, http://www.ebi.ac.uk/ena; submitted on 22 December 2022) and are available under accession number PRJEB56212. These data were analysed using the following bioinformatics pipeline. All reads were quality checked using FastQC v.0.11.9 [24], and processed using trimmomatic v.0.39 [25]. Paired-end reads from each *K. pneumoniae* isolate were assembled de novo using the Spades v.3.13.0 algorithm [26] to generate a draft genome sequence for each isolate. Quality assessment of the genome assemblies was performed using QUAST 5.0.2 [27]. In addition, the quality and completeness of the genome assemblies were confirmed using the BUSCO v.4.1.1 algorithm and the bacteria_odb10 dataset [28]

The PATRIC database [29] was used to identify complete or draft genomes of other *K. pneumoniae* strains isolated from patients in other sub-Saharan African countries. In-silico MLST typing was performed on all genomes by using the screen-Blast-mlst function of the Pathogenomics home-made R package (https://github.com/UCL-CTMA/Pathogenomics) and the *K. pneumoniae* allele sequences and MLST profiles available in the BIGSdb-Pasteur platform (https://bigsdb.pasteur.fr/klebsiella; accessed on 22 April 2022). It should be noted that the assignment of an allele type requires a perfect match with the corresponding sequence (i.e., 100% sequence identity and coverage). 

All genomes were submitted to kSNP3.0 [30] using the default settings, with the exception of a k-mer size of 19 and the ‘-core’ option. This software identified SNPs without the need for genome alignment or a reference genome. The phylogenomic tree based on the core genome was visualized using the R package ggtree [31]. The Pathogenomics custom R package was also used to screen all genomes for the virulence factor genes described in the VFDB database (http://www.mgc.ac.cn/VFs/; accessed on 24 August 2022 ) [32] and which are specifically associated to *K. pneumoniae*. The thresholds for sequence identity and coverage were respectively set at 80% and 50%. The R package ggtree was used to display the coverage percentage of each virulence gene in grayscale heatmaps. In silico K-typing and O-typing were performed on draft genomes of the 37 *K. pneumoniae* isolates, using Kleborate (v.2.2.0) [33] and Kaptive (v.2.0.3) [34]. Each isolate was assigned a virulence score based on the presence or absence of the yersiniabactin (*ybt*), colibactin (*clb)*, and aerobactin (*iuc*) genes, as previously described [33].

Simultaneously, each draft genome was screened for the presence of AMR genes and point mutations using the AMRFinderPlus 3.10.5 [35] tool developed by the National Center for Biotechnology Information (NCBI). The draft genomes were screened for the presence of plasmid incompatibility groups, in particular the IncF group. The Replicon Scheme Typing (RST) developed by Villa et al. [36] was used for classification of IncF plasmids according to the FAB (FII, FIA, FIB) formula. The DNA sequence of the novel FIA replicon was submitted to the curator of the pMLST database (https://pubmlst.org/plasmid/; accessed on 26 July 2022) for ST assignment.

## 3. Results

### 3.1. Bacterial Strains

A set of MDR ESBL-producing *K. pneumoniae* isolates were identified from the total *K. pneumoniae* (n = 89/449, ~20%) isolated from the GNB collection assembled between 2013 and 2017 at the Bukavu General Hospital, South Kivu, DRC. Due to isolate loss, either because of a problem with isolate storage or other technical issues, only 37 of the original 89 MDR ESBL-producing *K. pneumoniae* isolates were thoroughly analysed. The isolates were not duplicate clinical samples, but rather one sample per patient. The majority (n = 31, 83.8%) of the isolates were cultured from samples obtained from patients who were hospitalized and included all isolates from bloodstream (n = 19, 51.4%), urinary tract (n = 9, 24.3%), peritoneal fluid (n = 2, 5.4%), and soft tissue (n = 1, 2.7%). The remaining (16.2%) isolates (n = 6) were collected from patients with community-acquired urinary tract infections.

### 3.2. Antimicrobial Susceptibility Testing

All *K. pneumoniae* isolates (n = 37) were susceptible to imipenem, meropenem, and colistin while showing reduced susceptibility to ampicillin, amoxicillin, cefotaxime, ceftazidime, and ceftriaxone. While the majority of them (30 out of 37) showed reduced susceptibility to cefepime, they all exhibited the ESBL-production phenotype (Table 1 and Table 2).

### 3.3. Phylogenomic Analysis

SNP-based phylogenetic analysis of the MDR ESBL-producing *K. pneumoniae* from DRC revealed a high level of genetic diversity. The 37 *K. pneumoniae* isolates were distributed in 16 different clusters (Figure 1), each of which corresponded to a unique MLST profile (Table 3).

The most common sequence types (STs) were ST607 (n =6), ST48 (n = 6), ST340 (n = 5) and ST39 (n = 4). Other ST (i.e., ST15, ST874, ST1777 and ST37) had 2 isolates, while ST14, ST16, ST502, ST337, ST967, ST11, ST309 and ST2094 had only one. ST48 isolates (n = 6) were all from blood or peritoneal fluid, while ST340 isolates (n = 5) were obtained from blood (n = 4) or urinary tract (n = 1) from infected patients. ST607 isolates (n = 6) were obtained from patients suffering from bloodstream infection (n = 3) or urinary tract infection (n = 3). 

While the DRC lineages shared the same ST as *K. pneumoniae* isolates from several sub-Saharan African countries, their clustering and virulence gene content set them apart (Figure 1). Furthermore, 64.9% of DRC isolates shared the same ST as lineages found all over the world (e.g., ST14, ST15, ST39, ST48, ST340, and ST607). However, unlike the international clones which also belong to ST14 [37], ST15 [38], ST39 [39], ST48 [40], and ST340 [41,42], their DRC counterparts were susceptible to carbapenems and lacked carbapenemase gene. 

### 3.4. Genetic Determinants of Virulence

#### 3.4.1. Detection of Virulence Factors Genes

Among 128 virulence genes that were screened, 77 were found in at least one investigated isolate (Figure 1), with 27 of these virulence genes identified in each isolate. Table 3 summarizes data on ST, capsule typing, major virulence genes, and replicon typing. *K. pneumoniae* isolates harboured four distinct *ybt*: *ybt9*/conjugative integrative element (ICE) ICE*Kp3* (n = 6, among which 4 displayed a truncated ICE), *ybt14*/ICE*Kp15* (n = 12), *ybt14*/ICE*Kp12* (n = 2), *ybt15*/ ICE*Kp11* (n = 2), and *ybt16*/ICE*Kp12* (n = 4). The 37 *K. pneumoniae* isolates lacked the genes for colibactin (*clb*), salmochelin (*iroBCDN*), or aerobactin (*iucABCD-iutA*), as well as genes for hypervirulence (*rmpA*, *rmpA2* genes, and *peg-344*) [43,44]. The virulence score was either of 0 or 1, reflecting the absence or presence of the *ybt* gene. 

#### 3.4.2. Replicon Typing

According to the FAB formula, the distribution of the replicon types among 33 of the 37 *K. pneumoniae* isolates was F13:A13:B− (n = 9), F13:A−:B− (n = 7), F−:A13:B− (n = 5), F2:A13:B− (n = 4), and F1:A13:B− (n = 3), while only one isolate had the replicon type F1:A−:B−, F2:A−:B−, F8:A13:B−, F9:A−:B−, or F12:A26:B−. The FIA26, a novel replicon, was reported for the first time in one isolate, whereas replicon types could not be characterized in four *K. pneumoniae* isolates. Apart from IncF replicons, no other incompatibility plasmid replicon types (i.e., IncA/C, IncH1, IncH2, IncI1, or IncN) were found in *the K. pneumoniae* isolates.

#### 3.4.3. Identification of the Capsule Type 

In the 37 *K. pneumoniae* isolates, five different O types were assigned: O1 (37.8%), O2a (13.5%), O2afg (8.1%), O3b (13.5%), and O4 (8.1%). No O type could be determined in 18.9% isolates. Only three different K types could be assigned to a total of five isolates: K2 in isolates CTMA_1655 (ST39) and CTMA_1682 (ST39); K14 in isolates CTMA_1656 and CTMA_1660 (ST37); and K62 in isolate CTMA_1684 (ST48). The remaining thirty-two isolates could not be assigned a K-type with certainty. 

#### 3.4.4. Detection of AMR Genes

AMR determinants included resistance genes and point mutations (Figure 2 and Figure 3, respectively). The four MDR ESBL-producing *K. pneumoniae* isolates that lacked the CTX-M-15 gene were CTMA_1654 (ST309), CTMA_1656 and CTMA_1660 (ST37), and CTMA_1688 (ST11). However, CTMA_1654 and CTMA_1656 carried *bla_SHV-11_*, while CTMA_1660 and CTMA_1688 harboured *bla_SHV12_* and *bla_SHV_*, respectively. 

In addition, other *bla*-genes associated with ESBL-phenotype were widespread in all MDR ESBL-producing *K. pneumoniae* isolates. These genes included *bla_SHV_*_-1_, *bla_SHV_*_-28_, *bla_SHV-_*_62_, *bla_SHV_*_-110_, and *bla_SHV_*_-121_. We observed a redundancy of ESBL-production associated genes, with several isolates harbouring *bla_CTX_*_-*M*-15_ (n = 33), *bla_TEM_*_-1_ (n = 31), and/or the above-cited *bla_SHV_* genes (n = 37). It is noteworthy that no isolates simultaneously carried multiple *bla_SHV_* or *bla_TEM_*. Other β-lactamases genes included *bla_OXA_* (n = 13), *bla_OXA_*_-1_ (n = 14), and *bla_SCO_*_-1_ (n = 6) [45].

All the MDR ESBL-producing *K. pneumoniae* isolates carried the *fosA* gene associated with resistance to fosfomycin. All DRC *K. pneumoniae* isolates carried at least one gene linked to resistance to aminoglycosides and/or quinolones gene (i.e., *aac*(*3*)-*IId*, *aac*(3)-IIe, *aad*A1, *aad*A16, *aad*A2, *aph*(3′)-Ia*, aph*(3′’)-Ib, *aph(6)-Id*, *aac*(6′)-Ib-cr, or *aac*(6′)-Ib-cr5), as well as to phenicols (*catA1*, *catA2*, and *catB*), sulfonamides (*sul1* and *sul2*), tetracycline (*tet(*A), or *tet*(D)), and trimethoprim (*dfrA1*, *dfrA12*, *dfrA14*, *dfrA15*, *dfrA25*, and *dfrA27*). The two exceptions were the CTMA 1654 (ST309) and CTMA 1982 (ST2094) which lacked tetracycline resistance genes. 

Plasmid-mediated AMR to quinolone was found in 26 *K. pneumoniae* isolates, with plasmid-mediated quinolone resistance (PMQR) genes *qnr*B, *qnr*B1, *qnr*B2, *qnr*B6, *qnr*S1 genes and plasmid-mediated efflux pumps genes *oqx*A, *oqx*B, *oxq*B5, *oqx*B19, *oqx*B21, and/or *oqx*B32. 

#### 3.4.5. Detection of AMR Point Mutations 

In addition to AMR genes, AMR point mutations were identified (Figure 3). ST340 isolates (CTMA_1653, CTMA_1674, CTMA_1676, and 1678), and the ST11 isolate (CTMA_1688) carried the substitution S83I in the quinolone resistance-determining region (QRDR) of the gyrase A gene (*gyr*A), as well as the S80I substitution in the QRDR of the Deoxyribonucleic acid topoisomerase IV subunit A gene (*par*C). The two ST15 isolates CTMA_1657 and CTMA_1668 both had the S83F and D87A substitutions in *gyr*A, as well as the S80I substitution in the *par*C. 

In ST2094 (CTMA_1682) and ST16 (CTMA_1689) isolates, the *gyr*A gene harboured the S83F and D87N substitutions, while *par*C had the E84K or S80I substitution. Likewise, ST874 isolates CTMA_1663 and CTMA_1667 both harboured the S83Y and D87A substi-tutions in *gyr*A as well as the S80I substitution in *par*C. 

In the *gyr*A gene of five ST48 isolates, including CTMA_1650, CTMA_1661, CTMA_1662, CTMA_1666, and CTMA_1672, the S83Y substitution was the only mutation. The QRDR of *par*C from these isolates lacked any additional mutations.

Surprisingly, 9/37 (24,3%) MDR ESBL-producing *K. pneumoniae* isolates carried the colistin resistance-associated R256G substitution in the chromosomally encoded efflux pump *pmr*B gene. These isolates were typed ST340 (n = 5), ST37 (n = 2), ST11 (n = 1), and ST309 (n = 1). None of the *K. pneumoniae* isolates had any genes that were associated with reduced susceptibility to carbapenems.

## 4. Discussion

Being the cause of frequent outbreaks in hospitals, *K. pneumoniae* is becoming a significant global cause of severe infections. Despite reports of a high prevalence of MDR *K. pneumoniae* in a tertiary hospital in the eastern city of Bukavu in the DRC [16,17,22], neither this province nor the entire DRC have yet carried out molecular characterization of MDR ESBL-producing *K. pneumoniae*. We believe that this WGS study is the first to give insights on the genomic characteristics of MDR ESBL-producing *K. pneumoniae* isolates in this district of South-Kivu, eastern DRC. According to the WGS results, the MDR ESBL-producing *K. pneumoniae* isolates from clinical samples collected at the Bukavu hospital between 2013 and 2017 were found to have comparable ST to those commonly reported elsewhere.

The absence of carbapenemase genes, however, makes the current isolates significantly different from the global lineages, whose susceptibility to carbapenems is known to be decreased by these genetic determinants. A widespread use of carbapenems to treat severe ESBL-associated infection is likely too new in Bukavu to have seen the emergence of pandemic MDR ESBL-producing carbapenem-resistant *K. pneumoniae* (CRKP) lineages. This hypothesis is consistent with other data on MDR ESBL-producing *K. pneumoniae* in Ghana [46] and in southwestern Nigeria [47], where the administration of carbapenems was still restricted at the time of collection of isolates. In any case, the current results highlight the potential risk of a routine nontargeted administration of antimicrobial carbapenems, which is becoming more common in the province as a factor favouring the rapid emergence of carbapenem-resistant *K. pneumoniae* lineages. 

A second difference between the current eastern DRC *K. pneumoniae* isolates is the absence of STs, which are commonly found in other MDR ESBL-producing *K. pneumoniae* global clonal lineages, notably ST101 [20,48], ST147 [20,49], ST258 [50], and ST307 [20,49,51,52].

When compared to several genomes from other sub-Saharan African countries, the higher content of virulence genes in the current eastern DRC *K. pneumoniae* isolates is a third difference (Figure 1). It is worth noting, however, that no hypervirulence gene was found in the eastern DRC isolates. This result is consistent with other data describing nonhypervirulent MDR ESBL-producing *K. pneumoniae* in hospital-acquired infections. 

An intriguing finding is the presence of mutations associated with resistance to colistin in otherwise colistin-susceptible MDR ESBL-producing isolates. This observation was indeed made on isolates collected from patients whose treatment did not include colistin and in a province where there is no record of colistin administration. Regarding the use of colistin, these results should raise concerns about a potential rise of colistin-resistant MDR ESBL-producing *K. pneumoniae*, should this antibiotic be routinely introduced to treat infections. 

Our study had several limitations. First, the relatively small number of isolates (37 out of 89 possible) is a source of potential bias, as we might have overlooked several phenotypic (antimicrobial susceptibility) and genetic features (phylogenomics, virulence factors and AMR genes) of several potential lineages of MDR ESBL-producing *K. pneumoniae* in this tertiary care hospital. Secondly, several sequences including those on chromosome and genetic mobile elements can be missing from the draft genomes generated by Illumina short reads. Their use in subsequent genomic analyses can be potentially hampered by this incompleteness [53]. This is evidenced by the failure to assign any replicon to isolates carrying plasmid-mediated AMR genes (i.e., isolates CTMA_1659; CTMA_1664, and CTMA_1667). 

In conclusion, our study, despite its limitations, highlights the high genetic variability of MDR ESBL-producing *K. pneumoniae* in this easternmost province of DRC, calls for additional research across the whole country to determine the prevalence of these MDR ESBL-producing *K. pneumoniae* isolates, and suggests characterizing AMR genetic determinants, as well as virulence factor genes, that could account for their spread in the country and beyond. In practice, these results call for caution when administering colistin and carbapenems as well as parallel monitoring of the genomic and phenotypic evolution of AMR in DRC.

## Figures and Tables

**Figure 1 microorganisms-11-00525-f001:**
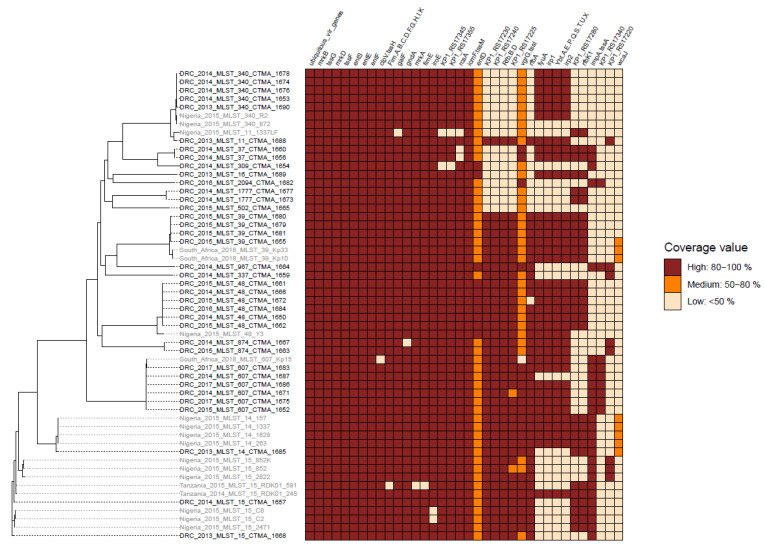
Phylogenetic tree of MDR ESBL-producing *K. pneumoniae* isolates from the DRC and other countries. The heatmap of *K. pneumoniae*-related virulence genes from the Virulence Factor Database (VFDB) is shown on the right side of the figure. The kSNP3 parsimony tree is based on single nucleotide polymorphisms (SNPs) within the core genomes of *K. pneumoniae* isolates from the DRC (in black) as well as isolates from sub-Saharan countries such as Nigeria, Tanzania, and South Africa (in grey). The heatmap’s red cells represent highly (>80%) covered genes, while the orange and yellow cells represent partially (50—80%) covered and uncovered (<50%) genes, respectively. Several genes, including *entD* and *vgrG*, have large deletions. *acrA*, *acrB*, *dotU*-*tssL*, *entA*, *entB*, *entC*, *fepA*, *fepB*, *fepC*, *fepD*, *fepG*, *fes*, *hcp*-*tssD*, *iutA1*, *mrkC* are examples of ubiquitous virulence genes.

**Figure 2 microorganisms-11-00525-f002:**
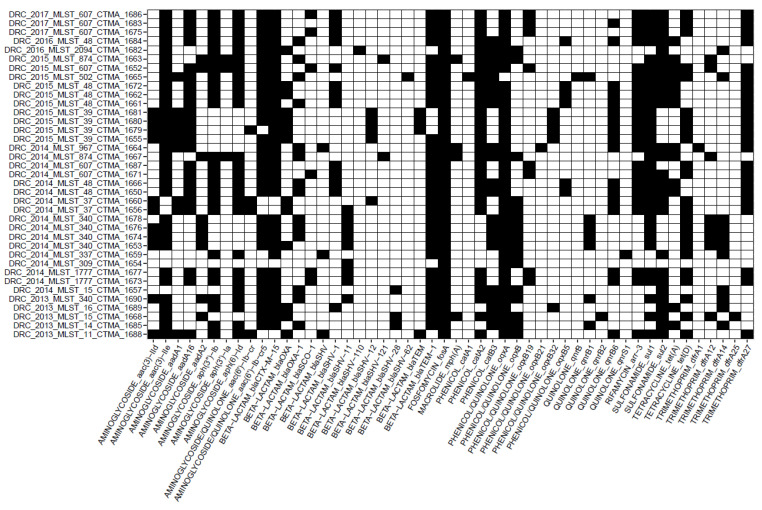
Heatmap of the genes involved in Antimicrobial Resistance (AMR) and identified with the AMRFinder tool. Black cells indicate the presence of the corresponding AMR gene.

**Figure 3 microorganisms-11-00525-f003:**
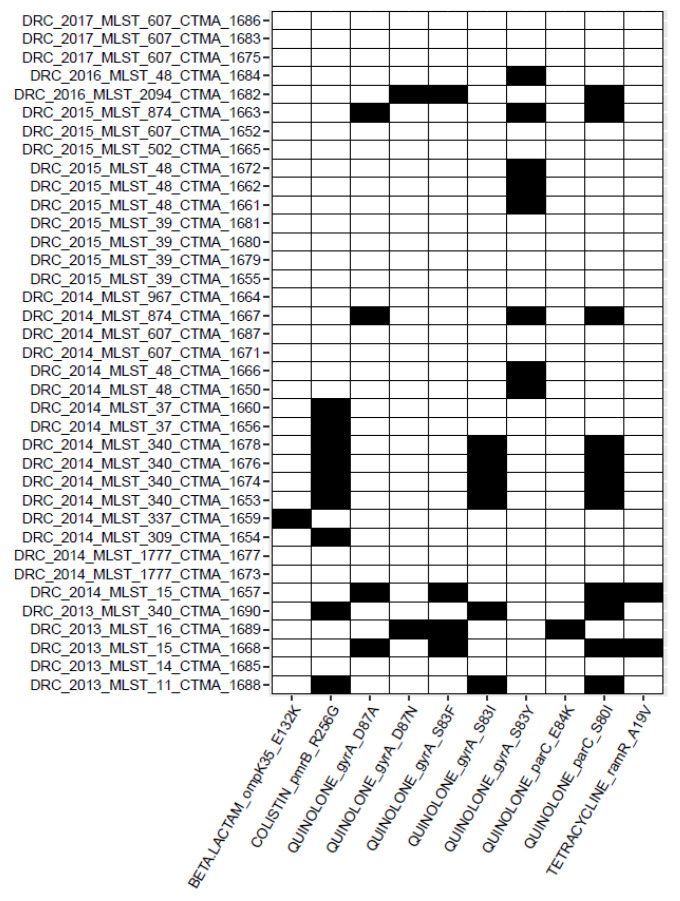
Heatmap of antimicrobial resistance (AMR) point mutations identified with the AMRFinder tool. Black cells indicate the presence of the corresponding SNP determinant.

**Table 1 microorganisms-11-00525-t001:** Antimicrobial susceptibility profiles of ESBL-producing *K. pneumoniae* isolates (n = 37) from the DRC. MIC values (mg/L) for tested antimicrobial agents.

Isolate	Tissue	ST	AMP	AMX	AMC	FEP	CTX	CAZ	CRO	IPM	MERO	CIP	AKN	GENT	TET	COL	NITRO
CTMA-1678	Bloodstream	340	>256	>256	>256	48	>256	>256	>32	0.094	0.75	>32	12	96	>256	0.25	>512
CTMA-1674	Bloodstream	340	>256	>256	>256	64	>256	>256	>32	0.094	0.094	>32	6	>256	>256	0.25	>512
CTMA-1676	Bloodstream	340	>256	>256	>256	64	>256	>256	>32	0.094	0.047	>32	8	64	>256	0.25	>512
CTMA-1653	Bloodstream	340	>256	>256	>256	32	>256	>256	>32	0.5	0.47	>32	8	64	>256	0.25	>512
CTMA-1690	UTI	340	>256	>256	>256	64	>256	192	>32	0.19	0.094	>32	8	128	>256	<0.25	>512
CTMA-1668	UTI	15	>256	>256	>256	48	>256	>256	>32	0.25	0.032	>32	6	96	>256	0.25	48
CTMA-1660	UTI	37	>256	>256	16	0.25	12	12	>32	0.125	0.023	1	24	>256	>256	<0.25	3
CTMA-1656	UTI	37	>256	>256	>256	0.75	24	24	>32	0.19	0.23	1	32	>256	>256	0.25	6
CTMA-1654	UTI	309	>256	>256	>256	12	>256	>256	>32	0.125	0.032	>32	24	>256	>256	0.25	64
CTMA-1689	UTI	16	>256	>256	>256	48	>256	>256	>32	0.25	0.002	>32	8	96	>256	0.25	>512
CTMA-1682	UTI	2094	>256	>256	>256	64	>256	>256	>32	0.023	0.125	>32	12	48	4	0.5	32
CTMA-1677	Bloodstream	1777	>256	>256	>256	12	64	16	>32	0.094	0.064	1.5	3	96	>256	0.25	>512
CTMA-1673	Bloodstream	1777	>256	>256	12	12	>256	>256	>32	0.25	0.047	1.5	3	96	>256	0.25	128
CTMA-1665	Peritoneal fluid	502	>256	>256	>256	24	>256	>256	>32	0.002	0.23	4	24	128	>256	<0.25	12
CTMA-1680	Bloodstream	39	>256	>256	>256	24	96	>256	>32	0.125	0.032	3	48	>256	>256	0.25	96
CTMA-1679	Bloodstream	39	>256	>256	>256	24	>256	>256	>32	0.125	0.064	2	48	>256	>256	0.25	32
CTMA-1681	Bloodstream	39	>256	>256	>256	32	>256	32	>32	0.125	0.032	2	64	>256	>256	0.5	96
CTMA-1655	UTI	39	>256	>256	>256	32	>256	>256	>32	0.19	0.32	2	24	>256	>256	0.25	6
CTMA-1664	UTI	967	>256	>256	8	8	>256	24	>32	0.064	0.47	1.5	8	24	>256	0.25	64
CTMA-1659	UTI	337	>256	>256	8	24	>256	16	>32	0.125	0.05	0.75	2	0.75	>256	0.25	24
CTMA-1661	Soft tissue	48	>256	>256	>256	16	>256	>256	>32	0.094	0.023	6	6	32	>256	<0.25	128
CTMA-1666	Bloodstream	48	>256	>256	64	8	>256	>256	>32	0.023	0.125	1.5	8	24	64	<0.25	96
CTMA-1672	Bloodstream	48	>256	>256	1.5	16	>256	>256	>32	0.19	0.047	>32	16	96	>256	0.25	>512
CTMA-1684	Bloodstream	48	>256	>256	>256	16	>256	>256	>32	0.094	0.047	8	12	32	16	0.5	128
CTMA-1650	Bloodstream	48	>256	>256	>256	16	>256	>256	>32	0.094	0.023	>32	16	96	>256	0.25	>512
CTMA-1662	Soft tissue	48	>256	>256	64	24	>256	>256	>32	0.094	0.023	6	6	32	>256	<0.25	128
CTMA-1667	Bloodstream	874	>256	>256	>256	1.5	>256	>256	>32	0.047	0.094	0.5	38	0.094	>256	0.25	16
CTMA-1663	Bloodstream	874	>256	>256	48	24	>256	>256	>32	0.5	0.047	>32	8	96	>256	0.25	192
CTMA-1683	Bloodstream	607	>256	>256	>256	16	>256	>256	>32	0.256	0.125	2	4	128	>256	0.25	>512
CTMA-1687	UTI	607	>256	>256	>256	>256	>256	>256	>32	0.094	0.064	1.000	3	>256	>256	0.5	>512
CTMA-1686	UTI	607	>256	>256	>256	>256	>256	>256	>32	0.19	0.023	0.094	6	192	1.5	0.5	>512
CTMA-1671	Bloodstream	607	>256	>256	>256	1.5	>256	>256	>32	0.094	0.047	>32.000	12	128	>256	0.25	>512
CTMA-1675	Bloodstream	607	>256	>256	>256	48	>256	>256	>32	0.125	0.047	0.125	4	>256	>256	0.25	192
CTMA-1652	UTI	607	>256	>256	>256	>256	>256	>256	>32	0.125	0.47	0.125	4	>256	>256	0.25	>512
CTMA-1685	UTI	14	>256	>256	>256	6	>256	>256	>32	0.19	0.023	1.5	6	96	128	0.5	>512
CTMA-1657	UTI	15	>256	>256	>256	16	>256	>256	>32	0.125	0.06	>32	8	64	>256	0.25	64
CTMA-1688	Bloodstream	11	>256	>256	>256	48	>256	>256	>32	0.256	0.023	1.500	6	64	2	0.25	48

Abbreviation: ST: Sequence Type; AKN: Amikacin; GENT: Gentamicin, AMP: Ampicillin; AMX: Amoxicillin; FEP: Cefepime; IMP: Imipenem; MERO: Meropenem; AMC: Amoxicillin/Clavulanic Acid; CAZ: Ceftazidime; CTX: Cefotaxime; CRO: Ceftriaxone; CIP: Ciprofloxacin; NITRO: Nitrofurantoin; COL: Colistin. R: Resistant, S: Susceptible, UTI: Urinary tract infection.

**Table 2 microorganisms-11-00525-t002:** AMR percentages of ESBL-producing *K. Pneumoniae* isolates according to CLSI/EUCAST guidelines.

Antibiotics	Resistance (R) n = 37 (%)	Susceptible (S) n = 37 (%)	Area of Technical Uncertainty (ATU) n = 37 (%)
Ampicillin	37 (100)	0 (0)	0 (0)
Amoxicillin	37 (100)	0 (0)	0 (0)
Amoxicillin-clavulanate	35 (94.6)	2 (5.4)	0 (0)
Cefepime	34 (91.9)	3 (8.1)	0 (0)
Cefotaxime	37 (100)	0 (0)	0 (0)
Ceftazidime	37 (100)	0 (0)	0 (0)
Ceftriaxone	37 (100)	0 (0)	0 (0)
Imipenem	0 (0)	37 (100)	0 (0)
Meropenem	0 (0)	37 (100)	0 (0)
Ciprofloxacin	35 (94.6)	1 (2.7)	1 (2.7)
Amikacin	15 (40.5)	22 (59.5)	0 (0)
Gentamicin	35 (94.6)	2 (5.4)	0 (0)
Tetracycline	34 (91.9)	3 (8.1)	0 (0)
Colistin	0 (0)	37 (100)	0 (0)
Nitrofurantoin	24 (64.9)	13 (35.1)	0 (0)

**Table 3 microorganisms-11-00525-t003:** MLST sequence type (ST), K and O and their respective K (capsule) and O antigen (LPS) type, as predicted by Kleborate.

Year of Isolation	Strain	Origin	ST	Replicon ST	K_Type	O Type	Yersiniabactin	(Hyper) Virulence Genes	Virulence Score
2014	CTMA_1678	Bloodstream	ST340	F1:A−:B−	unknown (KL107)	unknown (O4)	*ybt 9*; ICE*Kp3* (truncated)	-	1
2014	CTMA_1674	Bloodstream	ST340	F1:A13:B−	unknown (K15)	O4	*ybt 9*; ICE*Kp3* (truncated)	-	1
2014	CTMA_1676	Bloodstream	ST340	F1:A13:B−	unknown (K15)	O4	*ybt 9*; ICE*Kp3* (truncated)	-	1
2014	CTMA_1653	Bloodstream	ST340	F1:A13:B−	unknown (KL107)	unknown (O4)	*ybt 9*; ICE*Kp3* (truncated)	-	1
2013	CTMA_1690	UTI	ST340	F−:A13:B−	unknown (K15)	O4	*ybt 14*; ICE*Kp5*	-	1
2013	CTMA_1688	UTI	ST11	F13:A13:B−	unknown (KL105)	O2afg	*ybt 9*; ICE*Kp3*	-	1
2014	CTMA_1660	UTI	ST37	F−:A13:B−	K14	O3b	*ybt 15*; ICE*Kp11*	-	1
2014	CTMA_1656	UTI	ST37	F−:A13:B−	K14	O3b	*ybt 15*; ICE*Kp11*	-	1
2014	CTMA_1654	UTI	ST309	ND	unknown (K42)	unknown (O4)	-	-	0
2013	CTMA_1689	UTI	ST16	F12:A26:B−	unknown (K51)	O3b	*ybt 9*; ICE*Kp3*	-	1
2016	CTMA_1682	UTI	ST2094	F−:A13:B−	unknown (KL107)	unknown (OL101)	-	-	0
2014	CTMA_1677	Bloodstream	ST1777	F13:A13:B−	unknown (KL111)	O3b	-	-	0
2014	CTMA_1673	Bloodstream	ST1777	F13:A13:B−	unknown (KL111)	O3b	-	-	0
2015	CTMA_1665	Soft tissue	ST502	F13:A−:B−	unknown (K15)	unknown (O4)	-	-	0
2015	CTMA_1680	Bloodstream	ST39	F2:A13:B−	unknown (K2)	O2a	*ybt 16*; ICE*Kp12*	-	1
2015	CTMA_1679	Bloodstream	ST39	F2:A13:B−	unknown (K2)	O1	*ybt 16*; ICE*Kp12*	-	1
2015	CTMA_1681	Bloodstream	ST39	F2:A13:B−	unknown (K2)	O1	*ybt 16*; ICE*Kp12*	-	1
2015	CTMA_1655	UTI	ST39	F2:A13:B−	K2	O1	*ybt 16*; ICE*Kp12*	-	1
2014	CTMA_1664	UTI	ST967	F2:A−:B−	unknown (K18)	O2afg	-	-	0
2014	CTMA_1659	UTI	ST337	ND	unknown (KL109)	O2afg	-	-	0
2015	CTMA_1661	Peritoneal fluid	ST48	F13:A−:B−	unknown (K62)	O1	*ybt 14*; ICE*Kp5*	-	1
2014	CTMA_1666	Bloodstream	ST48	F13:A−:B−	unknown (K62)	O2a	*ybt 14*; ICE*Kp5*	-	1
2015	CTMA_1672	Bloodstream	ST48	F13:A−:B−	unknown (K62)	unknown (O2a)	*ybt 14*; ICE*Kp5*	-	1
2016	CTMA_1684	Bloodstream	ST48	F13:A−:B−	unknown (K62)	O1	*ybt 14*; ICE*Kp5*	-	1
2014	CTMA_1650	Bloodstream	ST48	F13:A−:B−	K62	O1	*ybt 14*; ICE*Kp5*	-	1
2015	CTMA_1662	Peritoneal fluid	ST48	F13:A−:B−	unknown (K62)	O1	*ybt 14*; ICE*Kp5*	-	1
2014	CTMA_1667	Bloodstream	ST874	ND	unknown (K45)	O1	*ybt 14*; ICE*Kp12*	-	1
2015	CTMA_1663	Bloodstream	ST874	ND	unknown (K45)	unknown (O1)	*ybt 14*; ICE*Kp12*	-	1
2017	CTMA_1683	Bloodstream	ST607	F13:A13:B−	unknown (K25)	O1	*ybt 14*; ICE*Kp5*	-	1
2014	CTMA_1687	UTI	ST607	F13:A13:B−	unknown (K25)	O1	-	-	0
2017	CTMA_1686	UTI	ST607	F13:A13:B−	unknown (K25)	O1	*ybt 14*; ICE*Kp5*	-	1
2014	CTMA_1671	Bloodstream	ST607	F13:A13:B−	unknown (K25)	O2a	*ybt 14*; ICE*Kp5*	-	1
2017	CTMA_1675	Bloodstream	ST607	F13:A13:B−	unknown (K25)	O1	*ybt 14*; ICE*Kp5*	-	1
2015	CTMA_1652	UTI	ST607	F13:A13:B−	unknown (K25)	O2a	ybt 14; ICE*Kp5*	-	1
2013	CTMA_1685	UTI	ST14	F9:A−:B−	K2	O1	-		0
2014	CTMA_1657	UTI	ST15	F8:A13:B−	unknown (KL112)	O1	-	-	0
2013	CTMA_1668	Bloodstream	ST15	F−:A13:B−	unknown (K48)	O2a	-	-	0

Abbreviation: UTI: Urinary tract infection. The Yersiniabactin column represents *ybt* genes alleles and their respective ICE*Kp*. The virulence genes (colibactin (*clb*), aerobactin (*iucABCD-iutA)*, salmochelin (*iroBCDN*)), as well as hypervirulence genes (*rmp*A, *rmp*A2 genes, and *peg-344*) were not identified in the 37 isolates.

## Data Availability

Publicly available datasets were analysed in this study. Raw sequencing data from all *K. pneumoniae* isolates were submitted to the European Nucleotide Archive (ENA, http://www.ebi.ac.uk/ena) and are available under accession number PRJEB56212 numbers. The data used and/or analysed during the current study are available from the corresponding author upon reasonable request. The DNA sequence of the novel FIA replicon was submitted to the curator of the pMLST database (https://pubmlst.org/plasmid/; submitted on 26 July 2022) for ST assignment.

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
