# Peer review of "Genomic Characterization of Multidrug-Resistant Extended Spectrum β-Lactamase-Producing Klebsiella pneumoniae from Clinical Samples of a Tertiary Hospital in South Kivu Province, Eastern Democratic Republic of Congo"

_microorganisms, 2023, doi:10.3390/microorganisms11020525_

Round 1
Reviewer 1 Report
The authors aimed to characterize extra-intestinal EBSL-producing K. pneumoniae clinical isolates using WGS recovered from the General Hospital of Bukavu, Congo.
Overall, the manuscript sounds good and is of great interest to readers.
However, I suggest creating a table on AMR percentages of Klebsiella isolates is also required. The antibiotics can be arranged according to the order of antibiotics/classes listed in the CLSI/EUCAST guidelines. You can show the percentage of isolates (%) resistant to the different antibiotic agents and to different antibiotic classes.
An additional table/figure describing the AMR profiles of Klebsiella isolates and/or showing the main AMR patterns will make your data more readable.
Author Response
REVIEWER N°1: Comments and Suggestions for Authors
The authors aimed to characterize extra-intestinal EBSL-producing K. pneumoniae clinical isolates using WGS recovered from the General Hospital of Bukavu, Congo. Overall, the manuscript sounds good and is of great interest to readers.
However, I suggest creating a table on AMR percentages of Klebsiella isolates is also required. The antibiotics can be arranged according to the order of antibiotics/classes listed in the CLSI/EUCAST guidelines. You can show the percentage of isolates (%) resistant to the different antibiotic agents and to different antibiotic classes.
An additional table/figure describing the AMR profiles of Klebsiella isolates and/or showing the main AMR patterns will make your data more readable.
Answer:
- The content of original table 1 has been arranged according to the order of antibiotics/classes listed in the CLSI/EUCAST guidelines, as requested.
- An additional Table 1b has been added, as suggested, to report AMR percentages of ESBL-producing Pneumoniae isolates.
Reviewer 2 Report
- Line 51-54, there are many studies on MDR of Klebsiella pneumoniae. At least 300 publications and 4131 BioProject of Klebsiella pneumoniae are available on NCBI.
- Line 135, since the MLST type is sensitive to mutation of sequences, what is the minimum sequence identity and coverage of the home-made package to determine the MLST type?
- Line 142-144, the VFDB database collects virulence genes from all species, did the authors use all the genes in VFDB or only genes from k. pneumoniae as reference when perform blasţ searching for virulence genes? Also, the authors should set a cut-off for coverage of sequence to increase reliability. Blast searching usually returns a lot of very short fragments with high identity. These fragment may only represent common structures shared by proteins from different families and should be removed from the results.
- Figure 1.
- Converts a black and white heat map to a color heat one.
- As mentioned above, treat low coverage VF genes as absence.
- Integrate the information such as Origin, ST, K/O type, etc. in Table 2 to Figure 1. Otherwise, the reader will repeatedly switch back and forth between Figure 1 and Table 1, which is a very unpleasant experience. This can be done with a web-based tool itol.
- Line 151-154, to my knowledge, AMRFinderPlus does not identify whether the gene is on the plasmid or on the chromosome. The authors needs other methods to determine if the resistance genes in Figure 2 and 3 on the plasmid or not.
Author Response
REVIEWER N°2: Comments and Suggestions for Authors
- Line 51-54, there are many studies on MDR of Klebsiella pneumoniae. At least 300 publications and 4131 BioProject of Klebsiella pneumoniae are available on NCBI.
Answer:
We intended to emphasise the scarcity of Klebsiella WGS data in Africa. The manuscript now includes this information.
- Line 135, since the MLST type is sensitive to mutation of sequences, what is the minimum sequence identity and coverage of the home-made package to determine the MLST type?
Answer:
This information is now specified in the manuscript
- Line 142-144, the VFDB database collects virulence genes from all species, did the authors use all the genes in VFDB or only genes from pneumoniae as reference when perform blasţ searching for virulence genes? Also, the authors should set a cut-off for coverage of sequence to increase reliability. Blast searching usually returns a lot of very short fragments with high identity. These fragments may only represent common structures shared by proteins from different families and should be removed from the results.
Answer:
The coverage threshold was set at 50%. This is now specified in the manuscript
- Figure 1.
- Converts a black and white heat map to a color heat one.
- As mentioned above, treat low coverage VF genes as absence.
- Integrate the information such as Origin, ST, K/O type, etc. in Table 2 to Figure 1. Otherwise, the reader will repeatedly switch back and forth between Figure 1 and Table 1, which is a very unpleasant experience. This can be done with a web-based tool itol.
Answer:
- We converted the black and white heatmap to a colour heatmap.
- VF genes with low coverage (<50%) were treated as absent. This is specified in the manuscript.
- Due to the very high number of categories associated with the variables in Table 2, We decided to keep Figure 1 and Table 2 separate. Due to the very high number of categories associated with the variables in Table2, importing so many data from Table 2 into Figure 1 would make the figure completely unreadable (including with itol). We decided to keep Figure 1 and Table 2 separate. However, we have taken note of your very pertinent remark and have improved the readability of the manuscript by presenting the classification of isolates in Table 2 in the same order as in Figure 1.
- Line 151-154, to my knowledge, AMRFinderPlus does not identify whether the gene is on the plasmid or on the chromosome. The authors needs other methods to determine if the resistance genes in Figure 2 and 3 on the plasmid or not.
Answer:
Thank you for your comment. Indeed, AMRFinderPlus does not allow to distinguish if the gene is located on a plasmid or on the bacterial chromosome (...unlike PointFinder which we used in previous studies on other bacterial species).
A solution to identify this location in our K. pneumoniae isolates is to produce a complete genome (e.g. by combining Illumina and ONT sequencing results). Although interesting for our current research on this type of isolates, such investigations are outside the scope of the current study.
In order to avoid potential confusion in the attribution of genes and point mutations to the chromosome and/or plasmid, we have modified the manuscript by keeping the distinction between AMR genes and point mutations.